# Impaired Mucosal Homeostasis in Short-Term Fiber Deprivation Is Due to Reduced Mucus Production Rather Than Overgrowth of Mucus-Degrading Bacteria

**DOI:** 10.3390/nu14183802

**Published:** 2022-09-15

**Authors:** Annelieke Overbeeke, Michaela Lang, Bela Hausmann, Margarete Watzka, Georgi Nikolov, Jasmin Schwarz, Gudrun Kohl, Kim De Paepe, Kevin Eislmayr, Thomas Decker, Andreas Richter, David Berry

**Affiliations:** 1Centre for Microbiology and Environmental Systems Science, University of Vienna, 1030 Vienna, Austria; 2Doctoral School in Microbiology and Environmental Science, University of Vienna, 1030 Vienna, Austria; 3Division of Gastroenterology and Hepatology, Department of Internal Medicine 3, Medical University of Vienna, 1090 Vienna, Austria; 4Joint Microbiome Facility of the Medical University of Vienna & the University of Vienna, 1030 Vienna, Austria; 5Department of Laboratory Medicine, Medical University of Vienna, 1090 Vienna, Austria; 6Department of Biotechnology, Faculty of Bioscience Engineering, Center for Microbial Ecology and Technology, Ghent University, 9000 Ghent, Belgium; 7Max Perutz Labs, University of Vienna, Vienna Biocenter (VBC), 1030 Vienna, Austria

**Keywords:** mucus secretion, quantitative microbiome profiling, fiber deficiency, intestinal shortening

## Abstract

The gut mucosal environment is key in host health; protecting against pathogens and providing a niche for beneficial bacteria, thereby facilitating a mutualistic balance between host and microbiome. Lack of dietary fiber results in erosion of the mucosal layer, suggested to be a result of increased mucus-degrading gut bacteria. This study aimed to use quantitative analyses to investigate the diet-induced imbalance of mucosal homeostasis. Seven days of fiber-deficiency affected intestinal anatomy and physiology, seen by reduced intestinal length and loss of the colonic crypt-structure. Moreover, the mucus layer was diminished, *muc2* expression decreased, and impaired mucus secretion was detected by stable isotope probing. Quantitative microbiome profiling of the gut microbiota showed a diet-induced reduction in bacterial load and decreased diversity across the intestinal tract, including taxa with fiber-degrading and butyrate-producing capabilities. Most importantly, there was little change in the absolute abundance of known mucus-degrading bacteria, although, due to the general loss of taxa, relative abundance would erroneously indicate an increase in mucus degraders. These findings underscore the importance of using quantitative methods in microbiome research, suggesting erosion of the mucus layer during fiber deprivation is due to diminished mucus production rather than overgrowth of mucus degraders.

## 1. Introduction

Nutrition can play a major role in health and disease. A healthy diet is considered to incorporate a variety of different nutritional sources such as nuts, fruits, legumes, and vegetables, whilst moderate in fish, poultry, and dairy products [1]. Many studies have shown that a variant to this, known as the Mediterranean diet, can extend lifespan, thus emphasizing the importance of our nutritional intake [2]. Optimal nutrition is somewhat individualized, as genetic predisposition can affect response to nutrition, and diet can in turn influence gene expression as well as the gut microbiome [3,4,5]. The gut microbiome itself can also influence health and response to dietary components, for example by microbial fermentation of fibers, as well as modifying fat accumulation and leptin sensitivity [6].

Fiber is an important component of a balanced diet, giving stool its consistency, delaying transit time, increasing water retention, and acting as an energy source for gut bacteria [7,8]. Additionally, fiber intake has been linked to lowered cholesterol levels as well as increased number of colonic goblet cells [9,10]. The recommended daily fiber intake is 14 g/1000 kcal, yet 85% of adults do not meet these recommendations [11]. This is problematic as fiber deficiency is related to cardiovascular diseases, colon cancer, and obesity [12,13,14,15,16].

The gastro-intestinal tract is lined with a mucosal layer consisting of secreted mucin glycoproteins produced by goblet cells, of which Muc2 is the main component in the intestine [10,17,18]. The mucosal layer protects colonocytes from pathogen invasion, microbial and dietary toxins, as well as antigens. Additionally, it represents a vital niche for microbes, which in return produce short-chain fatty acids (SCFAs) important for host health [18,19]. The main SCFAs produced in the gut are acetate, propionate, and butyrate. The latter, for example, has numerous beneficial functions, including being an energy source for epithelial cells, regulating mucus secretion, and having anti-inflammatory effects [19,20,21]. Propionate enhances satiety and reduces cholesterol, whereas acetate can be rerouted into butyrate and in itself can maintain intestinal immune homeostasis by stimulating intestinal IgA production [20,22,23,24]. A fiber-rich diet is imperative to support the synergistic relationship between the host mucosal environment and the gut microbiome [25,26].

A lack of fiber has been shown to cause erosion of the mucosal layer in gnotobiotic mouse models with low complexity gut microbiota [27,28]. Current literature proposes that a bloom of mucosal degraders during fiber deprivation drives this erosion [27]. However, as diet has been associated with altered goblet cell function, it remains unclear whether the host or the mucus-degrading microbiota is driving the imbalance in mucosal homeostasis. In this study we performed quantitative microbiome profiling, analysis of host mucosal anatomy and physiology, and stable isotope probing to determine compartment-specific changes in the host and intestinal microbiota. Our results indicate that the reduction of the mucosal layer is driven by impaired mucus production rather than by overgrowth of mucus degrading bacteria.

## 2. Materials and Methods

### 2.1. Animal Experiments

A total of 38 C57BL/6J mice (18 males, 20 females) were bred at the Centre for Biomedical Research, Medical University of Vienna, under specific pathogen-free (SPF) conditions and transferred to the Max Perutz Laboratories, University of Vienna, between 4–6 weeks of age. After an initial week of acclimatization on a standard chow control diet (CD, Ssniff, Soest, Germany), half the mice were switched to a fiber-free diet (FFD: without polysaccharides and cellulose, and with glucose as the only carbohydrate source; Ssniff, Soest, Germany) for another 7 days (Appendix A). The control diet contained numerous sources of polysaccharides such as wheat, soy, and barley. The remaining mice continued to receive the CD. Mice were co-housed yet split from the start by gender and diet in a controlled environment (14 h/10 h day/night cycle) with unlimited access to food and water. Two biological replicates were performed, yet not all methods were used for both experiments. Fecal pellets were collected on the morning of the diet switch, 2 days, and 7 days after switching the diet. Labeling was performed with stable isotope-labeled L-threonine (98 atom % ^13^C, 98 atom % ^15^N, Sigma-Aldrich, Burlington, MA, USA) in order to quantify mucus secretion. In 4 mice unlabeled L-threonine (Merck, Kenilworth, NJ, USA) was used as a ^12^C control. Compounds were dissolved in a sterile 0.9% NaCl solution and mice were injected with 50 µL via tail vein injection on day 7 as previously described [29]. Mice were sacrificed 6 h post-injection by cervical dislocation after which contents from cecum, small intestine, and colon were collected. Intestinal contents were split in two parts for both quantitative microbiome profiling and elemental and isotope analyses. A section of the colon was prepared for histological analyses. In the second replicate, the wet weight of the small intestine, cecum, and colon were measured with and without contents, to determine luminal content weight. In addition, the length of the small intestine and colon was measured.

### 2.2. Quantitative Microbiome Profiling

Flow cytometry was used to quantify the absolute number of bacterial cells per sample for small intestine, cecum, and fecal pellets using FACS Melody and the FACSChorus software (BD, Franklin Lakes, NJ, USA). Samples were dissolved in 500 µL PBS and then strained through a 40 µm cell strainer (Corning Inc., Corning, NY, USA). After filtering, 5 µL SYBR green was added which was later used for gating samples. Absolute counting beads (CountBright, Invitrogen, Thermo Fisher Scientific, Waltham, MA, USA) were used for cell counts according to the manufacturer’s instructions. Each sample was counted in triplicate and the mean total cell counts g^−1^ were calculated. DNA from intestinal contents was extracted using the FastStool DNA extraction kit automated on a QIACube Connect, following the manufacturer’s instructions. Thereafter, amplicons were generated using primers targeting the V4 region (515F Parada [5′-GTG YCA GCM GCC GCG GTA A-3′] and 806R Apprill [5′-GGA CTA CNV GGG TWT CTA AT-3′]) [30] of bacterial and archaeal 16S rRNA genes, barcoded in a unique dual setup, and sequenced on the Illumina MiSeq system as further described by Pjevac et al. [30]. Sequencing was performed at the Joint Microbiome Facility of the Medical University of Vienna and the University of Vienna under the project IDs JMF-1904-2 and JMF-2009-3. After sequencing, amplicon sequence variants (ASVs) were inferred [31] and classified following the analysis workflow detailed by Pjevac et al. [30]. Absolute taxon abundances, expressed as total counts per compartment, were derived from the flow cytometry cell counts in combination with the genus read table after copy number correction using *rrn*DB (5.7 pantaxa) based on Van de Putte et al. [32,33]. Sample load was calculated by summing the absolute abundances corrected by copy numbers. Load per compartment was calculated by multiplying the copy number corrected abundance by the total weight inside the intestinal compartment. It should be noted that the estimated abundances of rarer taxa are subject to larger measurement error associated with the sequencing depth and are therefore intrinsically of lower accuracy. To calculate the log2-fold change (log2FC) per diet, the top 50 genera where selected based on mean abundance. A pseudocount was added before taking the ratio of the FFD/CD. *t*-tests were used to evaluate significant statistical diet-induced abundance changes.

### 2.3. Elemental Analysis—Isotope-Ratio Mass Spectrometry

For elemental analysis—isotope-ratio mass spectrometry (EA-IRMS), 5 mg intestinal biomass was washed to eliminate free L-threonine. After washing, the pellet was dried overnight in a speedvac (Eppendorf Concentrator 5301, Eppendorf, Hamburg, Germany) and for another 24 h at 60 °C. These samples were used for wet- and dry-weight analysis. Next, 0.05–0.3 mg (dry weight) was transferred into a tin capsule. Samples for elemental analysis and ^13^C and ^15^N quantification were analyzed with an elemental analyzer (EA 1110, CE Instruments, Wigen, United Kingdom) coupled via a ConFlo III device to the IRMS (DeltaPLUS, Thermo Fisher) [29].

### 2.4. Mucosal Gene Expression Analysis

Epithelial cells were isolated according to the protocol by Gracz et al. [34]. After isolation, RNA was extracted using the RNAqueous-Micro Total RNA Isolation Kit (Thermo Fisher) following the manufacturer’s instructions. Subsequently, cDNA synthesis was performed using the High-Capacity cDNA Reverse Transcription Kit (Thermos Fischer). Real-time quantitative PCR was performed in a 20 µL reaction volume containing 1 µL template (cDNA diluted 1:5), 1× SYBR green Master Mix (Bio-Rad, Hercules, CA, USA), 7.4 µL H_2_O, and 0.8 µL of primers targeting either *muc2* (forward: 5′-ACTGCATGTGCGCGGCTCTT-3′, reverse: 5′-TGAGCTTGGGCAAGCGTGCA-3′) or the housekeeping gene 36B4 (forward: 5′-GCTTCATTGTGGGAGCAGACA-3′, reverse: 5′-CATGGTGTTCTTGCCCATCAG-3′). Amplification and detection were performed using a CFX96™ Real-Time PCR Detection System (Bio-Rad) using the following cycling conditions: 95 °C for 5 min, followed by 40 cycles of 95 °C for 15 s, 56 °C for 20 s, and 72 °C for 30 s. The concentration of cDNA of *muc2* was normalized to the concentration of the housekeeping gene 36B4. The ratio of normalized target concentrations (2-^ΔΔC^T method) was used to determine the fold change in gene expression [35].

### 2.5. Histological Analysis

Sections of the distal colon were fixed in 10% formalin overnight, after which samples were embedded in paraffin. Using a microtome, 4 µm thick sections were cut. Before staining with Alcian Blue, slides were heated at 60 °C for 30 min followed by two 5-min baths in xylene. Next, slides were submerged into 100% ethanol twice for 1 min, after which they were rehydrated in 96%, 80%, and 70% ethanol and water, each for 1 min. Slides were then incubated with Alcian Blue for 30 min at room temperature and washed with water to remove excess stain. After air-drying, Vectashield Hardset™ Antifade Mounting Medium (Vector Laboratories, Newark, CA, USA) was applied together with a cover slip and hardened at 4 °C overnight. Slides were scanned with a 40× objective (137.766 nm/pixel, SLIDEVIEW VS200, Olympus, Tokyo, Japan). ImageJ [36] software was used for crypt length measurements and staining intensity quantification. 

### 2.6. Statistical Analysis

Statistical analyses and data visualization were performed in R (version 4.1.1, R Core Team, Vienna, Austria) [37], using the R packages data.table (version 1.14.0, Matt Dowle, Bozeman, MT, USA) [38], rstatix (version 0.7.0, Alboukadel Kassambara, Marseille, France) [39], and ggplot2 (version 3.3.5, Hadley Wickham, Houston, TX, USA) [40]. The differences between diet and day were tested using a 2-way analysis of variance (ANOVA) for weight change, nutritional intake and alpha diversity, and with a 2-way ANOVA between diet and compartment for all other analyses, followed by Student’s *t*-test with a Bonferroni adjustment for significant factors. Prior to ANOVA, data was tested for normality with Shapiro–Wilk test and Levene’s test was used to test homoscedasticity. A square root transformation was applied when data was not normally distributed. For the sequencing data, samples were subsetted for a minimum of 1000 reads and rarified to a richness of 1763, after which alpha (observed OTUs, Shannon diversity, and Simpson index) and beta diversities (principal coordinates analysis [PCoA] with Bray–Curtis dissimilarity) were calculated using the R package vegan (version 2.5-7, Jari Oksanen, Helsinki, Finland) [41] and visualized using ampvis2 (version 2.7.9, Mads Albertsen, Aalborg, Denmark) [42]. Vegan was used both for permutational multivariate analysis of variance (PERMANOVA) calculations and for the Kendall coefficient of concordance [41]. Silhouette scores were determined to deduce the optimal number of clusters for identification of concordant clusters of bacterial taxa, by Ward clustering based on Spearman correlations [43].

## 3. Results

### 3.1. FFD Alters Intestinal Anatomy and Decreases Colonic Mucus Production

In order to investigate the role of acute dietary fiber deprivation on mucus secretion and utilization by the gut microbiota, mice were fed either normal chow as a control diet (CD, composition in Appendix A) or a fiber-free diet (FFD, composition in Appendix A) lacking cellulose and polysaccharides (two biological replicates; Appendix A). Fecal samples were taken immediately before, 2 days, and 7 days after diet switch, at which point mice were sacrificed and intestinal compartments were sampled. Food intake was not significantly different between groups for either diet or day of the dietary intervention (ANOVA, *p* = 0.170 and *p* = 0.224), and although FFD-fed mice had, on average, roughly 9000 joules per day higher energy intake (ANOVA, R^2^ = 0.37, *p* = 0.043), the overall gain in body weight was not affected by diet (ANOVA, *p* = 0.527; Appendix A). Notably, however, FFD induced a significant shortening of both the small intestine (CD: 36.1 ± 1.0, FFD: 34.1 ± 1.1 cm; *t-*test, *p* = 0.007) and the colon (CD: 8.9 ± 0.5 cm, FFD: 7.3 ± 0.6 cm; *t*-test, *p* < 0.001; Figure 1A). Consistent with this shortening, there was a marked reduction in the weight of the small intestine, cecum, and colon tissue (emptied of luminal contents) in FFD mice (*t*-test, *p* = 0.01, *p* = 0.01 and *p* < 0.001; Figure 1B).

To further explore FFD-induced changes in mucosal physiology, tissue structure and mucus production were evaluated in colon samples at day 7 (representative images of colon sections are shown in Figure 1C,D). There was a trend towards loss of large-scale tissue structure in FFD mice, with a slight, though not statistically significant, reduction in the number of intestinal longitudinal folds per colon section (*t*-test, *p* = 0.115; Figure 1E). Crypt height, as measured from the base to the top of the crypt, was significantly reduced in FFD mice (*t*-test, *p* < 0.001; Figure 1F). Although the number of mucus-containing goblet cells, as determined by Alcian Blue staining, was not affected by diet (*t*-test, *p* = 0.371; Figure 1G), the expression of the *muc2* gene, which encodes the most abundant secreted mucin in the colon, was greatly reduced in FFD mice (*t*-test, *p* = 0.002; Figure 1H). These results indicate that acute dietary fiber deprivation greatly impacts intestinal anatomy and colonic mucus production.

### 3.2. FFD Reduces Luminal Contents and Mucus Secretion

We next evaluated how FFD affects the luminal environment. Water content, as expected, decreased from proximal to distal compartments regardless of diet (ANOVA, R^2^ = 0.48, *p* < 0.001), but also water content in the colon was significantly lower in FFD-fed mice (*t*-test, *p* = 0.038; Figure 2A). The wet weight of luminal contents was reduced in FFD mice (ANOVA, R^2^ = 0.36, *p* < 0.001; Figure 2B), though the dry weight was not affected by diet (ANOVA, *p* = 0.477; Figure 2C). These differences likely reflect the water-retaining capacity of fiber and suggest a decreased nutrient absorption capacity which may result from the shortened intestines in FFD mice and is consistent with the unaffected body weight gain despite the higher energy intake by FFD-fed mice. Next, the percentage of total carbon and nitrogen in the luminal contents was used to establish a carbon to nitrogen ratio (C:N). The C:N ratio is an important measure of nutrient limitation in ecosystems, as N levels in the gut can affect bacterial load [44]. FFD-fed mice had a lower C:N ratio in each intestinal compartment (ANOVA, R^2^ = 0.18, *p* < 0.001; Figure 2D), which could largely be attributed to a reduction in the relative carbon content of the luminal biomass (%C; ANOVA, R^2^ = 0.49, *p* < 0.001; Figure 2E). This may be due to the lack of recalcitrant carbon-rich fiber as well as an alteration in the ratio of assimilable carbon and nitrogen-containing nutrients. In support of the concept of reduced assimilable nutrient levels, quantification of total bacterial load using flow cytometry indicated that there were fewer bacteria per intestinal compartment in FFD-fed mice (ANOVA, R^2^ = 0.15, *p* = 0.007; Figure 2F). Overall, these data suggest that fiber deficiency results in a marked reduction in overall luminal wet weight contents and microbial load.

To determine if the FFD-induced anatomical and physiological changes in mucosal tissue affect the amount of mucus secreted into the lumen, secretion was quantified using a previously-established stable isotope probing approach employing an intravenously-administered ^13^C^15^N threonine tracer [29]. Since Muc2 has a protein domain rich in threonine, proline, and serine, isotopically labeled threonine will be incorporated into the mucin glycoprotein and can thus be used as a read-out for secretion. Mucus secretion was detectable in all compartments, but the overall flux of secreted compounds (calculated by multiplying the amount of ^13^C with the total amount of carbon and the dry weight of the intestinal contents) per compartment was significantly lower for FFD-fed mice (ANOVA, R^2^ = 0.13, *p* = 0.041; Figure 2G). This result is in line with the reduced *muc2* gene expression in colonic tissue and indicates impaired mucus secretion due to fiber deficiency.

### 3.3. FFD Affects Gut Microbiota Composition

We next evaluated how FFD affects gut microbiota composition using 16S rRNA gene amplicon sequencing combined with flow cytometry to determine absolute abundances of bacterial taxa per intestinal compartment [33]. As differences in wet weight were observed between diets, it was decided to calculate abundances based on the weight of the entire intestinal compartment rather than the abundance per gram of sample. Changes in volume of intestinal contents would then be reflected in total bacterial load. There was a reduction in bacterial alpha diversity (observed OTU, Shannon diversity, and Simpson index) in stool samples of fiber-deficient compared to control mice collected 2 and 7 days after the start of the dietary intervention (ANOVA; R^2^ = 0.3, R^2^ = 0.34 and R^2^ = 0.32, *p* < 0.001 for all measures; Figure 3A). Principal coordinates analysis (PCoA) of Bray–Curtis dissimilarities showed a clustering of day 2 and 7 stool from FFD mice distinct from baseline (Figure 3B), and permutational multivariate analysis of variance supported that diet was a significant driver of stool microbiota composition (PERMANOVA; R^2^ = 0.15, *p* < 0.001). Notably, there was no statistically-significant difference in microbiota composition between days 2 and 7 (PERMANOVA, *p* = 0.73), indicating that an alternative stable microbiota state had been rapidly induced by FFD. Consistent with the stool diversity, the bacterial alpha diversity in the small intestine, cecum, and colon compartments also significantly decreased at day 7 in FFD-fed mice (ANOVA, R^2^ = 0.37, R^2^ = 0.61 and R^2^ = 0.56, *p* < 0.001 for all measures; Figure 3C). Samples in the PCoA showed clustering by diet and incomplete clustering by intestinal compartment (Figure 3D), although PERMANOVA testing revealed that both diet and compartment were significantly associated with microbiota composition (PERMANOVA; diet: R^2^ = 0.18, *p* = 0.001, compartment: R^2^ = 0.15, *p* = 0.002). The taxonomic composition of the intestinal microbiota was typical of previous reports of murine microbiota [4,45], and the most abundant genera included members of the *Bacteroidota* (*Muribaculum*, *Duncaniella*, *Paramuribaculum*, *Alistipes*, and *Odoribacter*), *Bacillota* (*Faecalibaculum*, *Intestinomonas*, and *Lactobacillus*), *Verrucomicrobia* (*Akkermansia*), and *Pseudomonadota* (*Desulfovibrio*) (Figure 3E). 

To determine which taxa were altered in their abundance due to FFD, we calculated the mean log2-fold change of the top 50 genera with respect to diet in each intestinal compartment (FFD/CD). This was done for both absolute abundances as well as relative abundances to facilitate comparison to previous studies [27,46,47]. In the small intestine, no genera were significantly altered by diet based on absolute or relative abundances (Figure 4A). In the cecum, 31 taxa were significantly decreased in their absolute abundance such as *Muribaculum, Duncaniella*, *Alistipes*, and *Paramuribaculum,* whereas in relative abundance data 37 taxa were significantly altered—both increased and decreased (Figure 4B). Notably, *Akkermansia*, which has previously been reported to bloom in FFD, was in fact not changed in absolute numbers, although it displayed a 1.47 Log-Base-2-fold increase in relative abundance. In the colon, 20 taxa were different in their absolute abundance, where 28 were shifted in their relative abundance, again including *Akkermansia* (Figure 4C). As expected, most of the genera affected by diet belonged to the *Bacillota*, including butyrate producers such as *Kineotrix* and *Clostridium,* as well as fiber degrading *Bacteroidota* such as *Muribaculum* [21,48,49]. These results highlight the issue with inferring population dynamics using relative abundance-based methods and indicate that *Akkermansia* does not bloom during dietary deprivation, but rather maintains its population size as other taxa are lost. 

Taxa that co-vary in abundance across samples may have similar environmental niche preferences, be involved in similar metabolisms, and/or be interacting with one another [50], and can be considered to be an ecological guild [51]. Co-variances in this dataset would be expected to be largely driven by host diet, yet could also arise from other factors such as inter-host variability. Concordance analysis indicated that there was indeed concordance of bacterial genera (Kendall concordance test; *p* < 0.001 for all compartments). To identify clusters of concordant genera, Ward clustering based on pairwise Spearman correlations of absolute abundances was performed. Silhouette scores indicated that genera could be optimally clustered into two groups in each compartment, and that cluster membership varied by compartment (Figure 4A–C). In the small intestine, diet did not influence the absolute abundance of either cluster which corresponds to no significant differences found in the genera (Figure 4D). In the cecum, however, cluster Ce1, which was the numerically-dominant group of bacteria in CD, was dramatically reduced in FFD (*t*-test, *p* = 0.01). Cluster Ce2, which included *Akkermansia*, *Faecalibacilum*, unclassified *Bacteroidales*, and *Desulfovibrio*, was not significantly affected by diet (*t*-test, *p* = 0.77), although as observed for the individual taxon dynamics, relative abundance analysis gave contradictory results (Figure 4E). A similar trend was observed in the colon, with the dominant cluster Col1 decreased in FFD (*t*-test, *p* = 0.048) and the less abundant cluster containing *Akkermansia* unchanged (*t*-test, *p* = 0.3, Figure 4E). These results provide evidence that dietary fiber deprivation does not induce a bloom of mucus degraders but instead induces a dramatic loss of non-mucus-degrading bacterial diversity. 

## 4. Discussion

Modern diets are increasingly lacking dietary fiber and studies have shown that numerous diseases have a direct link to a reduced fiber intake [12,13,14,15,16]. Lack of fiber has been linked to a diminished mucosal layer, yet thus far, the processes behind the depletion of the mucus layer remain poorly understood [25,27,28]. By using quantitative microbiome profiling and stable isotope probing, this study reveals that mucus layer depletion upon fiber deprivation is not microbiota-mediated but stems from a direct host response downregulating mucus production. Besides mucus production, fiber deficiency affected many other host and microbiota endpoints.

Plant materials contain many more components in addition to fiber that can have bioactivity for the microbiome and the host. It should be considered, that with the elimination of fiber from the diet, polyphenols as well as other plant components are also removed and could contribute to the observed effects. Additionally, that the replacement of fiber with glucose may potentially influence the results, which is a limitation in our study as well as others [25,26,27].

Fiber deficiency dramatically affects the intestinal structure, decreasing crypt height and shortening the length of both the small and large intestines (Figure 1). An important function of the intestines is to absorb nutrients, thus when intestines are shortened there is less surface area to do so, less area for goblet cells, and less area to form crypts. The intestinal shortening corroborates previous work by Desai et al. and Hunt et al., which show shortening after long-term fiber-deficient diets (40 and 21 + 122 days) as well as 1- and 4-day oscillations in diet [18,20]. Our work shows that this effect already occurs after 7 days. Intestinal shortening due to a fiber-deficient diet as well as a high fat and fiber-deficient diet has been found to be microbiota-independent, suggesting that this is not due to microbial factors such as lack of butyrate [26,27]. The gut harbors many intestinal hormones, some of which are influenced by diet. GLP-1 secretion is stimulated by fiber [52] and GLP-2 has been shown to play an important role in intestinal size, whereas, when acting together, they promote intestinal healing [53,54]. Knockout of GLP-1 and GLP-2 receptors in a mouse model, however, was insufficient to counteract fiber deficiency-induced intestinal shortening, suggesting the presence of another not-yet identified mechanism [46]. 

Elemental and isotope analysis were used to study changes in the elemental stoichiometry of the gut biomass and mucus secretion in vivo. EA-IRMS results showed that there was a lower percentage of total carbon in the gut lumen in FFD-fed mice (Figure 2), most likely from the lack of dietary fibers. The total wet weight of the intestinal lumen content was also decreased in FFD-fed mice (Figure 2). An important function of fiber is to retain water in the colon to increase stool bulk [7], which explains the decreased water content on the FFD seen for the colon yet not for the small intestine or cecum. The water content of mucus is 98% [18], thus less secreted mucus might partially explain the decreased water content. There was a decreased flux of secreted mucus into the gut lumen in FFD-fed mice, as determined using a ^13^C^15^N threonine tracer as a proxy for intestinal mucus secretion [29], as well as a reduced expression of the *muc2* gene, which encodes the major secreted mucin in the colon (Figure 2). These data indicate that fiber deficiency leads to a decrease in mucus production rather than an increase in microbial mucus degradation activity, as previously implied, supported by unchanged abundances of mucus degraders [27]. In accordance with these results, thinning and increased permeability of the mucus layer as well as reduction in colonic goblet cell numbers due to fiber deficiency have been previously observed in multiple studies [25,26,27]. This is further supported by the observation that mucus degrading taxa, including *Akkermansia muciniphila*, unclassified *Bacteroidales*, and *Desulfovibrio* did not increase in absolute abundance in fiber-deprived mice.

In contrast to the mucus-degrading specialists, other taxa quantitatively decreased upon fiber deprivation as reflected by the dramatically reduced alpha diversity of the intestinal microbiota across all compartments, which is in line with previous reports (Figure 3) [25,27,46,55]. Beta diversity showed a clear clustering based on diet (Figure 3). Clustering per compartment was evident for the CD but incomplete for the FFD. Low et al. also found that beta diversity and microbial community per compartment was altered by their high-fat and fiber-deficient diet, although they observed clustering to be more apparent on the intervention diet compared to the control diet [47]. Absolute abundance analysis showed that fiber degraders and butyrate producers such as *Lachnospiraceae* and *Muribaculum* were lost on the FFD whereas mucus degraders were not influenced by diet (Figure 4). Loss of butyrate-producing bacteria may be detrimental as butyrate is an energy source for colonocytes and plays an important role in gut health [56]. However, butyrate has been found not to affect *muc2* expression unless glucose is restricted [57], and therefore the loss of butyrate producers may not be the reason for the decrease in *muc2* expression observed in this study.

The loss of abundant fiber-associated taxa is likely the reason that an increase in mucus degraders during fiber deprivation has previously been reported [27,47]. Our results, however, showed that such compositional microbiome analysis greatly skews the interpretation of population dynamics and underscores the importance of quantitative microbiome profiling [33].

## 5. Conclusions

In conclusion, diet, and in particular fiber, plays a vital role in host health and establishment of the gut microbial community, and thus it is crucial to understand its direct and indirect effects on the host as well as the microbiota. Murine studies reversing fiber-deficient diets with normal chow, or transplanting gut microbiota from chow-fed mice to FFD-fed mice, show that the microbial community largely shifts back to a normal composition, whereas the intestines remained shortened, and the thickness of the mucus layer diminished for a prolonged period [26,47]. Thus, microbial shifts are acute yet reversible, whereas physiological changes and damage thereof are more long-lasting. Potential interventions should thus not target the inhibition of mucus degraders, but rather the promotion of fiber-degrading butyrate producers. 

## Figures and Tables

**Figure 1 nutrients-14-03802-f001:**
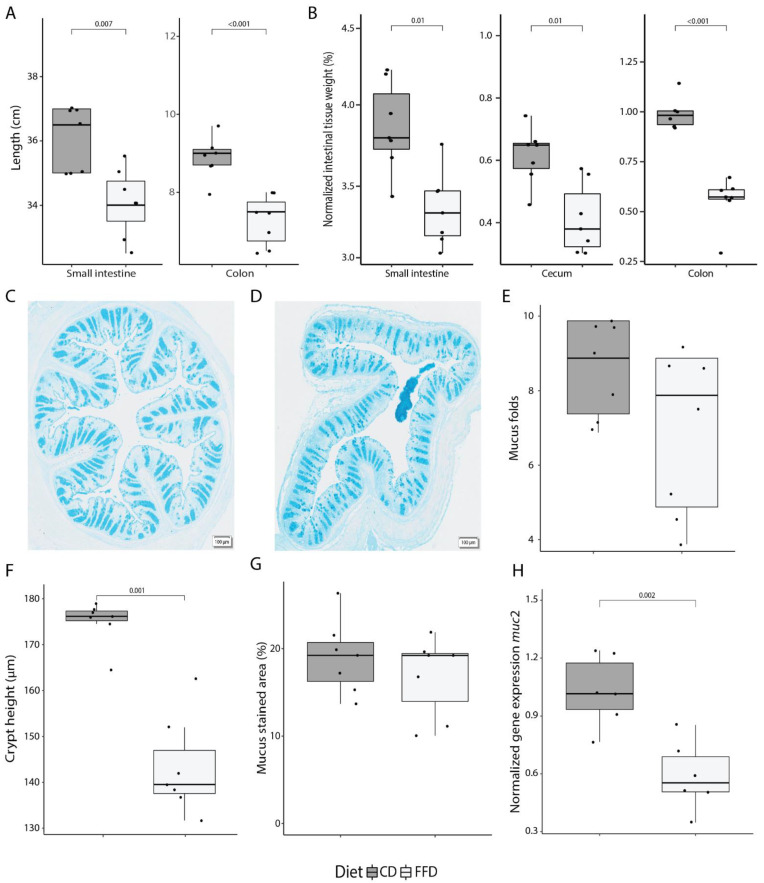
Dietary fiber deficiency affects intestinal anatomy and colonic mucus production. (**A**) Length of the small and large intestine is significantly decreased after 7 days of fiber deficiency (ANOVA, *p* < 0.001). (**B**) Fiber deficiency decreases intestinal tissue weight (ANOVA, *p* < 0.001). Values were normalized to the total body weight of the mouse. (**C**) A representative colon cross-section of a control mouse, and (**D**) fiber-deficient mouse displaying a loss in tissue structure at 40× magnification. (**E**) Using a cross-section of the colon, intestinal longitudinal folds were counted. A decreasing trend was observed in fiber-deficient mice. (**F**) Crypt height, as measured from the bottom to the top of the crypt, significantly decreased with fiber deficiency (*t*-test, *p* < 0.001). (**G**) Quantification of Alcian Blue, which stains acidic mucus and is an indirect measure for mucus-containing goblet cells, did not show a diet-dependent difference in the colonic cross-sections. (**H**) *muc2* expression, as quantified by qPCR, was significantly decreased with a fiber-deficient diet (*t*-test, *p* = 0.002). Box plots show the group median and interquartile range, with each dot representing a single sample. Dark grey indicates the control diet, light grey the fiber-free diet.

**Figure 2 nutrients-14-03802-f002:**
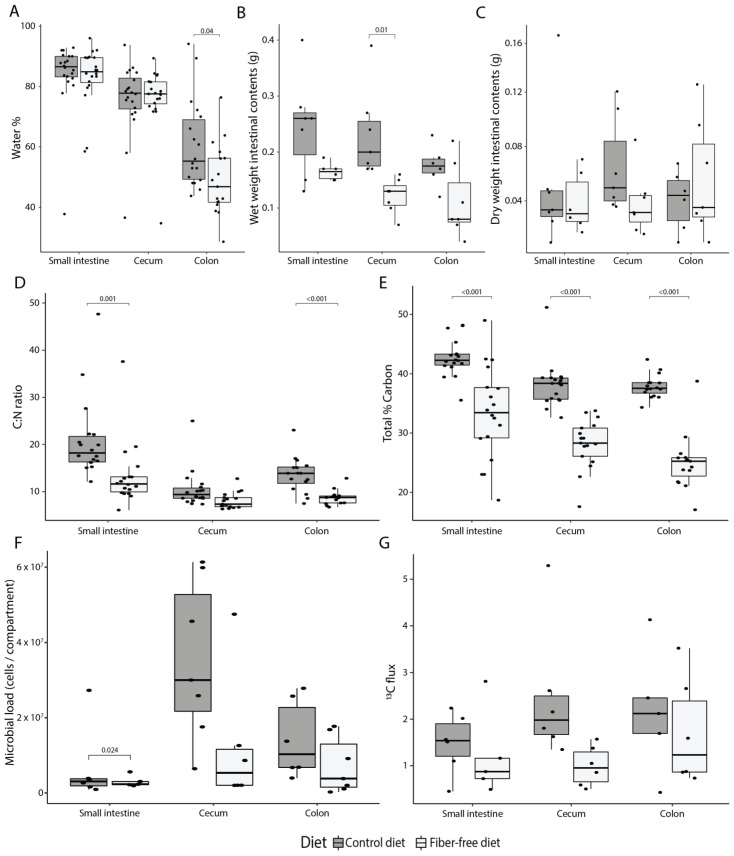
Effect of fiber deficiency on luminal contents and mucus secretion. (**A**) Percentage of water in the intestinal content decreases along the intestinal tract (ANOVA, *p* < 0.001), and is significantly less in the colon of mice fed a fiber-free diet (FFD) compared to a control diet (CD). (**B**) Fiber deficiency reduces the wet weight of intestinal contents (ANOVA, *p* < 0.001). (**C**) Dry weight of intestinal contents is not affected by diet. (**D**) Fiber deficiency significantly reduces the carbon to nitrogen ratio (ANOVA, *p* < 0.001). (**E**) Total percent of carbon in the lumen is significantly decreased with fiber deficiency (ANOVA, *p* < 0.001). (**F**) Fiber deficiency decreases the total amount of bacteria per compartment (ANOVA, *p* = 0.007). (**G**) Host-secreted carbon flux—calculated by multiplying the amount of ^13^C with the total amount of carbon and the dry weight of the intestinal contents—was significantly lower in FFD-fed mice (ANOVA, *p* = 0.041). Box plots show the group median and interquartile range, with each dot representing a single sample. Dark grey indicates the control diet, light grey the fiber-free diet.

**Figure 3 nutrients-14-03802-f003:**
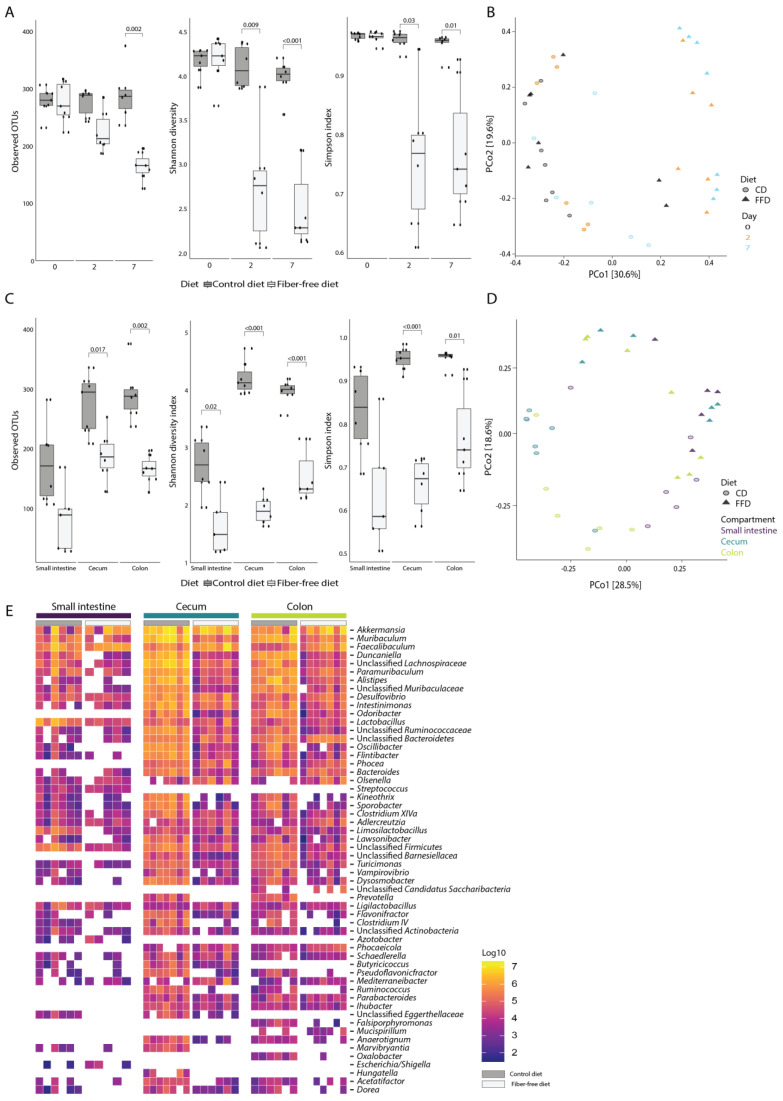
Dietary fiber deficiency decreases microbial alpha-diversity and is a key driver of microbial beta-diversity. Fecal samples were taken 0, 2, and 7 days after dietary intervention. (**A**) Fiber deficiency decreases bacterial richness and alpha diversity (ANOVA, *p* < 0.001 for all measures) in fecal samples taken 0, 2, and 7 days after dietary intervention. (**B**) Principal coordinates analysis based on Bray–Curtis dissimilarities shows a significant separation of samples based on diet (PERMANOVA; *p* < 0.001). (**C**) After 7 days, fiber deficiency significantly reduces bacterial richness and alpha diversity in all three intestinal compartments (ANOVA, *p* < 0.001 for all measures). (**D**) Principal coordinates analysis based on Bray–Curtis dissimilarities at genus level shows a significant separation by diet (PERMANOVA *p* = 0.001) and incomplete clustering by compartment (PERMANOVA *p* = 0.002). (**E**) Absolute genus abundances, ranked from highest to lowest, separated by diet and intestinal compartment (the 50 most abundant genera are shown). Box plots show the group median and interquartile range, with each dot representing a single sample. Dark grey indicates the control diet, light grey the fiber-free diet.

**Figure 4 nutrients-14-03802-f004:**
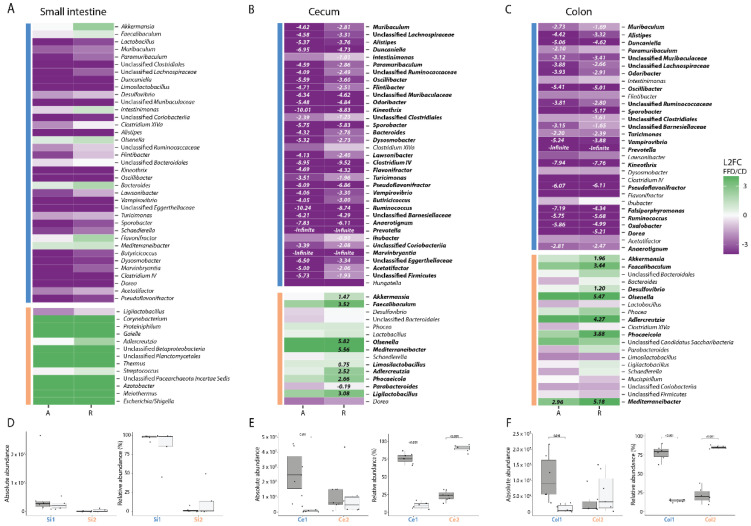
Dietary fiber deprivation induces significant shifts in the composition of the intestinal microbiota. (**A**–**C**) Log-base-2-fold change (L2FC) of each genus with respect to different diets (fiber-deficient/control) in each intestinal compartment, for both absolute (A) and relative (R) abundances. Taxa with significant L2FC values are printed and the L2FC value is given. Taxa are split by concordance group (i.e., cluster) and ranked from highest to lowest mean abundance. The absolute and relative abundances of taxa are grouped by cluster, diet, and compartment and displayed in box plots showing the group median and interquartile range, with each dot representing a single sample. Dark grey indicates the control diet, light grey the fiber-free diet. (**A**) + (**D**) for the small intestine, (**B**) + (**E**) for the cecum, (**C**) + (**F**) for the colon.

## Data Availability

The 16S rRNA gene amplicon sequencing data has been deposited at the Sequence Read Archive (https://www.ncbi.nlm.nih.gov/sra accessed on 26 January 2022) under the BioProject accession ID PRJNA800211. FACS data can be found in the flow repository (http://flowrepository.org/ accessed on 28 January 2022) with the ID FR-FCM-Z4UJ.

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
