# Peer review of "Impaired Mucosal Homeostasis in Short-Term Fiber Deprivation Is Due to Reduced Mucus Production Rather Than Overgrowth of Mucus-Degrading Bacteria"

_nutrients, 2022, doi:10.3390/nu14183802_

Round 1

Reviewer 1 Report

In the present study the authors demonstrated in the animal model that seven day fibre free diet affects negatively the anatomy and physiology of GI tract and has a negative impact on the composition of gut microbiota. Quantitative microbiome profiling of the gut microbiota showed diet-induced reduction in the gut microbiota diversity and decrease in SCFA-producing taxa. This study support the observation found in humans on fibre poor diet.

Author Response

We thank the reviewer for the positive assessment of our work.

Reviewer 2 Report

Impaired mucosal homeostasis in short-term fiber deprivation 2 is due to reduced mucus production rather than overgrowth of 3 mucus-degrading bacteria

First of all, I would like to thank the editor and the authors for allowing me to review this article. The authors evaluated the role of dietary fiber in maintenance of gut mucosal homeostasis and found that the erosion of the mucus layer during fiber deprivation is due to diminished mucus production rather than overgrowth of mucus degraders such as Akkermansia muciniphila. The theme and result are interesting. I think it will be useful for readers. To make this paper better, I have a few comments.

Major

1)      According to Supplementary Table 1, the difference between CD and FFD is not only starch and glucose. For example, FFD contains twice more over crude fat than in CD. Additionally, as the authors pointed out, CD contains many plant materials. In addition to polysaccharide, plants contain many kinds of ingredients, for example polyphenols, which have been reported many biological activities as well as the impact of gut microbiota. Since those ingredients have been removed in FFD, I think their impact may not be negligible. And composition of FFD possibly seems to be high glucose diet. What is the difference between fiber free diet and high glucose diet? Ideally, the diet composition should be similar between CD and FFD, with the exception of dietary fiber. But if it is technically difficult, the authors should discuss the differences, especially the ingredients which have been reported the impact of gut microbiome or mucus layer.

2)      There are two types of dietary fiber; soluble fiber and insoluble fiber, and they play a different role in the intestine. Insoluble dietary fiber, such as cellulose, affects gut transit via increasing the stool size. Using FFD induced constipation rat model, it has been reported that cellulose affects stool property including stool number, size, and weight [1]. If FFD mice showed the constipation phenotype, stool volume may be considered to include for absolute abundance analysis. And that report also reported that there was no association between cellulose and alpha-diversity. The author can discuss this point.

1.                 Oshiro, T.; Harada, Y.; Kubota, K.; Sadatomi, D.; Sekine, H.; Nishiyama, M.; Fujitsuka, N. Associations between intestinal microbiota, fecal properties, and dietary fiber conditions: The Japanese traditional medicine Junchoto ameliorates dietary fiber deficit-induced constipation with F/B ratio alteration in rats. Biomed. Pharmacother. 2022, 152, 113263, doi:10.1016/j.biopha.2022.113263.

Minor

3)       Figure 2 was too large to show evelything. So I couldn’t fully review in section 3.2.

4)       Evaluating microbiome using absolute abunance is excellent. On the other hand, the minimum sequence read number 1000 seems to be too small. I wonder if the lower ranked genera in top 50 seem to have risk of overestimation or underestimation in the absolute number analysis.
